# The Effects of Selected Measurement Errors on Surface Texture Parameters

**DOI:** 10.3390/ma15144758

**Published:** 2022-07-07

**Authors:** Pawel Pawlus, Rafal Reizer, Michal Wieczorowski, Wieslaw Zelasko

**Affiliations:** 1Faculty of Mechanical Engineering and Aeronautics, Rzeszow University of Technology, Powstancow Warszawy 8 Street, 35-959 Rzeszow, Poland; ppawlus@prz.edu.pl; 2Institute of Materials Engineering, College of Natural Sciences, University of Rzeszow, Pigonia Street 1, 35-310 Rzeszow, Poland; rreizer@ur.edu.pl; 3Division of Metrology and Measurement Systems, Faculty of Mechanical Engineering, Institute of Mechanical Technology, Poznan University of Technology, Piotrowo Street 3, 60-965 Poznan, Poland; 4Faculty of Mechanics and Technology, Rzeszow University of Technology, Kwiatkowskiego Street 4, 37-450 Stalowa Wola, Poland; w.zelasko@prz.edu.pl

**Keywords:** surface texture, measurement errors, parameters

## Abstract

Surface texture measurement, characterized by areal parameters, is very susceptible to measurement errors. Various types of errors differently affect surface texture parameters. In this paper, two types of measurement errors were investigated. To analyze the impact of the presence of scratches, circular valleys of various diameters were added to surface textures measured by a white light interferometer. Measurement errors were larger for higher scratches. Skewness and kurtosis were mostly affected by the presence in valleys, and changes of spatial parameters were also comparatively high. The difference between the results of measurement of the same surface details two times after a break of three months was also studied. This difference was caused by errors of relocation, spikes and non-measured point presences, high-frequency noise, and surface ageing. Spatial parameters were found to be the most stable.

## 1. Introduction

The surface texture is the fingerprint of the manufacturing processes. It affects the functional properties of machined elements, such as contact, sealing, friction, wear, and lubrication [1,2]. The introduction of three-dimensional measurement led to better analysis of surface topographies. The areal surface texture parameters allow a more correct analysis of surface features, such as example peaks [3]. The analysis of 2D profiles caused errors in the study of surface texture properties such as slope. However, surface texture measurement is very susceptible to measurement errors. These errors differently affected various areal surface texture parameters [4]. However, the impact of measurement errors has not yet been fully analyzed. Generally, measuring equipment, measurement procedure, interpretation of the results of measurement, and measured surface cause uncertainty of surface texture measurement.

Some of the errors in surface topography measurement are related to measurement methods. The errors typical to the stylus method are related to the co-action stylus tip and the surface texture. There are errors caused by mechanical filtration of the stylus tip related to trouble with penetration of the stylus tip into the valleys [5,6,7]. The other errors are related to stylus flight—the possibility of losing contact between stylus tip and surface texture, when the measurement speed is too high [8,9,10,11]. The errors typical of using optical methods are related to the presence of non-measured points [12] or spikes [13,14], stitching can be another source of errors [15,16]. High-frequency noise can also cause errors of surface topography measurement using optical and tactile methods [17]. Some errors are also related to temperature issues [18].

Regardless of the measurement type, errors can be related to the digitization process: quantization errors [19] and errors caused by sampling interval selection. The selection of the sampling interval should also be related to surface function [20,21,22]. Errors can be related to surface analysis. Filtration is the first action in the analysis of surface texture. There are problems with the filtering of two-process surfaces, such as plateau honed surfaces. The commonly used Gaussian filter can cause a large distortion of the roughness of such textures [23,24]. The calculations of parameters can be an additional source of errors. The measured surface is the other source of errors. It causes variation of parameters [25,26,27]. Rotation of the surface of the small angle affected the values of the surface texture parameters [28].

There are several parameters that describe the textures of the areal surface. They can be divided into height, spatial, hybrid, functional, end, and other. Various groups of parameters are designed for various surfaces, depending on the function of the surface [29,30]. However, parameters of different types are susceptible to various errors of surface topography measurement.

Although uncertainty of surface texture measurement was previously analyzed, the effects of errors on the values of surface texture parameters have not yet been studied. In particular, the impacts of the presence of additional valleys and the repetitive surface texture measurement by optical methods have not been studied. This work tries to fill this gap. The results of this work will be helpful for the selection of parameters characterizing surface texture, leading to an improvement in the quality of surface texture assessment. Variations in the values of surface texture parameters should be minimized.

## 2. Materials and Methods

Figure 1 presents a flowchart of measurements and analyses of surface textures.

The scratches can be the result of measurement errors, especially after using optical methods. The defects can be created during machining. The presence of scratches affects the values of the areal surface parameters. The first part of this study is related to this problem. Several surface topographies were measured by a white light interferometer Talysurf CCI Lite of 0.01 nm vertical resolution, using an objective of 5×. The sampling area of 3.29 mm × 3.29 mm contained 1024 data points. Flat surfaces were leveled and forms of curved surfaces were removed using the polynomial of the second degree. Digital filtration was not used. The parameters of the ISO 25178-2 standard [29] were calculated. These parameters are frequently used to characterize areal surface texture. Before computing the parameters, spikes were eliminated. Surfaces after honing, milling, polishing, lapping, and grinding were the objects of investigation. To find the sensitivity of parameters to the presence of scratches, circular valleys of defined radii were added to the surfaces. These valleys did not affect the maximum surface height. The distortions were filled in with the ordinates of the lowest point of the surface. Circular valleys had diameters of 0.1 or 0.15 mm.

In the second part of this study, surface textures of several surfaces were measured by a white light interferometer two times after a break of three months. The variability of the parameters of these surfaces depended mainly on the quality of relocation, spikes and the presence of non-measured points, high-frequency noise, and slight on surface ageing. The changes of the parameters of the areal surface were studied. Surfaces after various treatments, such as grinding, lapping, abrasive blasting, and abrasive blasting followed by lapping, were measured and analyzed. Surface textures were analyzed similar to the first part of this study. The methods of measurement and analysis were the same for both measurements. The surfaces were relocated using first mechanical and then digital methods, with the help of TalyMap 6 software.

Surfaces after various processing operations were studied to obtain diversity of textures. Surfaces after grinding are random one-directional structures. Milling led to the creation of deterministic surfaces. Random isotropic rough surfaces are created in abrasive blasting. Lapping and especially polishing led to the creation of smooth random surfaces. Surfaces after milling, grinding, vapour blasting, lapping, and polishing are one-process textures, which means that they have traces of only one machining process. However, plateau honing and abrasive blasting followed by lapping led to the formation of highly skewed two-process random textures. Surfaces after honing are anisotropic cross-hatched, surfaces after abrasive blasting and then lapping are isotropic. 

## 3. Results and Discussion

### 3.1. Impact of the Presence of Additional Valley

Figure 2 presents contour plots, material ratio curves, and selected parameters from the plateau-honed cylinder surface, without and with additional deep valleys. An increase in the width of the valley (diameter) caused an increase in the relative errors of the parameter calculation. Among the Sk group [31], the reduced dale height Svk increased by up to 5%, and the Sr1 material ratio (of small functional significance [29]) decreased to 9%, the changes in the other parameters were smaller. Among the amplitude parameters, the highest changes of RMS height Sq occurred (up to 3%). Hybrid parameters, the RMS slope Sdq, and the developed interfacial areal ratio Sdr were stable. The relative changes in the spatial parameters were large. The autocorrelation length Sal increased to 13% and the texture aspect ratio Str increased to 43%. Skewness Ssk decreased by up to 10%, and kurtosis Sku increased by up to 12%, and the extreme height of the peak Sxp increased by 3%. Skewness and kurtosis are frequently used to describe the shape of the ordinate distribution in surface description and modeling. The emptiness coefficient Sp/Sz (Sp—maximum peak height, Sz—maximum height of the surface) and Sq/Sa (Sa—arithmetical mean of the absolute surface heights) were found to be an alternative to skewness Ssk and kurtosis Sku [29]. The sensitivity of these parameters to the existence of an additional valley was small: the Sq/Sa increased to a maximum of 5% and changes in the Sp/Sz parameter were negligible.

Figure 3 presents contour plots, material ratio curves, and selected parameters from the milled surface, without and with additional deep valleys. Among the Sk family, only increases in the Svk parameter occurred up to 5%. Relative changes in amplitude, spatial parameters, and Sxp were smaller than observed for the plateau-honed surface; skewness and kurtosis were also small, up to 2%. The highest relative increase occurred for RMS slope Sdq, up to 11%. Similar high changes of this parameter took place for other analyzed deterministic surfaces, with a comparative long correlation length after turning or milling. The largest increase in the Sdr parameter was smaller, up to 4%. The parameters Sq/Sa and Sp/Sz were more stable than kurtosis Sku and skewness Ssk.

Figure 4 presents contour plots, material ratio curves, and selected parameters from a polished surface, without and with additional deep valleys. Among the parameters from the Sk group, the parameters Svk and Sr1 changed to 4%. The Ssk parameter decreased to 7%. The changes in other parameters were small. Sq/Sa and Sp/Sz were more stable than Sku and Ssk.

Figure 5 presents contour plots, material ratio curves, and selected parameters from the surface after abrasive blasting, without and with additional deep valleys. The Sq parameter increased to 2%. Among the Sk group, the Svk parameter increased to 4%. The Ssk parameter decreased to 13% and Sku increased to 7%. The Str parameter increased by up to 4%. Changes in other parameters were low. Similar to other analyzed surfaces, Sq/Sa and Sp/Sz were more stable than Sku and Ssk.

Figure 6 presents contour plots, material ratio curves, and selected parameters from the ground surface, without and with additional deep valleys. The Svk parameter increased to 4%. Changes in other parameters characterizing amplitude were negligible. The Ssk parameter decreased to 11% and Sku increased to 5%. The changes in spatial parameters were comparatively large. The Sal parameter increased to 7%, and the Str parameter increased to 5%. The Sxp parameter increased to 3%. The parameters characterizing the shape of the ordinate distribution Sq/Sa and Sp/Sz had a smaller sensibility to the presence of circular valleys than Sku and Ssk.

Changes in parameters depended on the radius of the valley. When it was larger, the changes in parameters were also greater. The parameters Ssk and Sku were characterized by the largest changes. The Ssk parameter decreased, while the Sku parameter increased. The relative changes in these parameters depended on the shape of the ordinate distribution. For surfaces characterized by negative skewness, these relative changes were higher compared to those of surfaces of symmetric ordinate distribution. Changes in the spatial parameters Sal and Str were also typically high. The Sal parameter of the anisotropic random surfaces increased. Different changes could occur for isotropic textures. For isotropic surfaces, the Str parameter decreased, in contrast to anisotropic surfaces. Relative changes in these parameters were larger for lower values. The parameters characterizing the part of the valley Svk (and the dale void volume Vvv) increased due to the presence of an individual valley (changes in Svk values were typically higher than those of Vvv). Changes in other parameters from the Sk family (core height Sk, the reduced peak height Spk, and Sr2 material ratio) and from the V group (the core void volume Vvc, peak material volume Vmp, and core material volume Vmc) were smaller. The averaged parameters Sa and Sq and hybrid parameters increased; changes in Sq were higher than those in Sa. The Sxp parameter increased. From definition, the Sz parameter was constant. The hybrid parameters Sdq and Sdr as well as Spc increased, and changes were small on random surfaces. High changes of slope Sdq occurred for deterministic surfaces with a comparatively large correlation length after turning or milling. It was found that the maximum valley depth Sv, the Sp parameter, the ten-point height S10z, the five-point peak height S5p, the five-point pit height S5v, the inverse material ratio Smc, and the peak density Spd were rather stable. The areal material ratio Smr, the average area of the hill Sha, and the average hill volume Shv were constant. The feature parameters, average dale area Sda and average dale volume Sdv, increased. In general, changes depended on the character of the surface. The sensitivity of Sp/Sz Sq/Sa on the existence of an additional valley was found to be much smaller than that of skewness and kurtosis for both skewed and non-skewed surfaces: the Sq/Sa increased by maximum 5% and changes in the Sp/Sz parameter were negligible.

### 3.2. Impact of Repetitive Measurement

Figure 7 presents contour plots, material ratio curves, and selected parameters from ground surface measured and re-measured. Among the parameters from the Sk group, the highest relative difference occurred for the Spk parameter (approximately 7%). The parameters characterizing the average height (Sa, Sq) were more stable than the parameters describing the maximum amplitude (Sp, Sv, Sz; the highest relative change was approximately 12%. The relative change in skewness was comparatively high (more than 20%), the kurtosis Sku was more stable (the relative change was approximately 7%). The variation of Sq/Sa was smaller than that of Sku and the variation of Sp/Sz was less than that of Ssk. Spatial parameters Sal and Str were constant. Relative changes in hybrid parameters Sdq and Sdr as well as Spc were small. The Smr and the Spd parameters changed by nearly 20%.

Figure 8 presents contour plots, material ratio curves, and selected parameters from a measured and re-measured lapped surface. Parameters from the Sk family were comparatively stable, the highest relative difference was approximately 3%. The average height parameters were more stable than the parameters describing the maximum amplitude (the highest deviation was greater than 10%). The parameters that describe the shape of the ordinate distribution were rather stable (the relative deviations were less than 5%). Although Sq/Sa was more stable than Sku, the variation of skewness of Ssk was less than that of Sp/Sz. Similar to the surfaces presented above, the spatial parameters Sal and Str were constant. The hybrid parameters Sdq and especially Sdr (more than 10%) changed as well as Spc. The changes in Spd and Smr parameters were high (more than 35%).

Figure 9 presents contour plots, material ratio curves, and selected parameters from the surface after measured and re-measured abrasive blasting. Similar to the surfaces analyzed above, parameters from the Sk group, as well as Sa and Sq parameters, were stable. The variation of the Sp parameter was close to 15%. Spatial parameters Sal and Str were constant. Ssk and Sku were stable. Changes in hybrid parameters Sdq and Sdr and of peak density Spd were small (up to 2.5%). However, the relative changes of the mean peak curvature Spc were greater than 30%. Sq/Sa was more stable than Sku and Ssk was more stable than Sp/Sz. The changes in the Smr parameter were very high.

Figure 10 presents contour plots, material ratio curves, and selected parameters from surface after abrasive blasting followed by measured and re-measured lapping. Similar to the surfaces analyzed above, parameters from the Sk group, as well as Sa and Sq parameters, were stable, and relative changes were smaller than 3%. The changes in the parameters describing the maximum height were higher, approximately 29%. Skewness Ssk changed by almost 20%, but kurtosis Sku changed by approximately 40%. The parameters Sp/Sz and Sq/Sa were more stable than Ssk and Sku. Changes in spatial parameters were smaller than 4%. The hybrid parameters and Spc changed, and the highest variation occurred for Sdr. The peak density Spd changed by more than 40%. The Smr changes were the largest.

Figure 11 presents contour plots, material ratio curves, and selected parameters from the surface after measured and re-measured grinding. The amplitude of this surface was much smaller compared to the surface shown in Figure 6. Among the Sk family, the highest changes occurred for the Svk parameter (more than 15%). The variations in the average amplitude parameters were much smaller than those of the parameters describing the maximum height; from these parameters, Sv changed the most (more than 15%). The Ssk parameter changed greatly (near 40%), and the variation of kurtosis Sku was smaller (near 15%). The parameters Sp/Sz and Sq/Sa were more stable than those of Ssk and Sku, respectively. Similar to most of the analyzed surfaces, the spatial parameters Sal and Str were constant. The relative change of the Sdr parameter was two times greater than that of Sdq, and it amounted to 25%. However, the Spc parameter was rather stable. The Smr parameter changed by 40% and peak density changed by 25%.

It was found that averaged amplitude parameters such as Sa and Sq were more stable on surfaces than parameters characterizing maximum height such as Sp, Sv, and Sz. The Smr parameter was characterized by very high variability. The Smc parameter was more variable than Sxp. The parameters in the V group were more stable than the parameters in the Sk family. The spatial parameters Sal and Str were very stable. Variations in Ssk and Sku parameters were typically high. The Sq/Sa ratio was more stable than Sku, and in most cases the emptiness coefficient Sp/Sz was more stable than Ssk. The variations of the Spc parameter and of the hybrid parameters Sdq and Sdr were sometimes comparatively high. Among these parameters, the Spc parameter typically had the highest variation, followed by Sdr and Sdq. The feature parameters, especially Spd, Sda, Sha, Sdv, and Shv, were characterized by high variability.

## 4. Conclusions

Various parameters react differently on the measurement errors. Parameter changes depend on the type of surface.Changes in parameters due to the presence of an additional valley depended on the radius of the valley. When it was larger, these changes were also larger. The presence of scratches caused changes in amplitude parameters characterizing the valley parts Svk and Vvv; the changes in the other amplitude parameters from the Sk and the V groups were smaller. The parameters Ssk and Sku were characterized by large changes. Changes in the spatial parameters Sal and Str were also typically high. The hybrid parameters changed substantially for deterministic surfaces of a large wavelength.Skewness and kurtosis are frequently used to describe the shape of the ordinate distribution. The pair of parameters: Sp/Sz and Sq/Sa is the alternative to skewness Ssk and kurtosis Sku. It was found that the sensitivity of Sp/Sz and Sq/Sa to the existence of an additional valley was much lower than that of skewness and kurtosis.After measurements of the same surfaces and after a break of a few months, variation of surface texture parameters occurred. This difference can be restricted by the use of the proper relocation method and the same measurement procedure.Repetitive measurements caused large variations of most of the feature parameters (Spd, Sda, Sha, Sdv, and Shv) as well as Smr, Ssk, and Sku parameters. The average amplitude of parameters was more stable than those that characterized the maximum surface height. The spatial parameters Sal and Str were very stable. The variation of Sq/Sa was smaller than that of Sku.Independently of the type of error, the parameters from the V group were more stable than the parameters from the Sk family.The results of this research can be used for the selection of parameters characterizing surface textures of various types. This will lead to an improvement in the quality of industry surface texture assessment.The findings are limited to surfaces measured by optical methods. In the future, impacts of other measurement errors will be studied.

## Figures and Tables

**Figure 1 materials-15-04758-f001:**
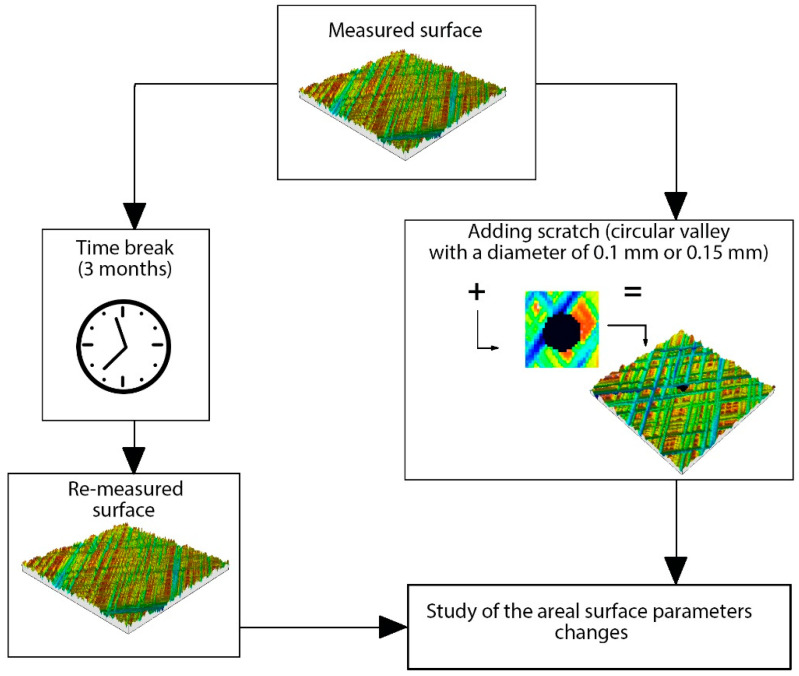
Flowchart of research work.

**Figure 2 materials-15-04758-f002:**
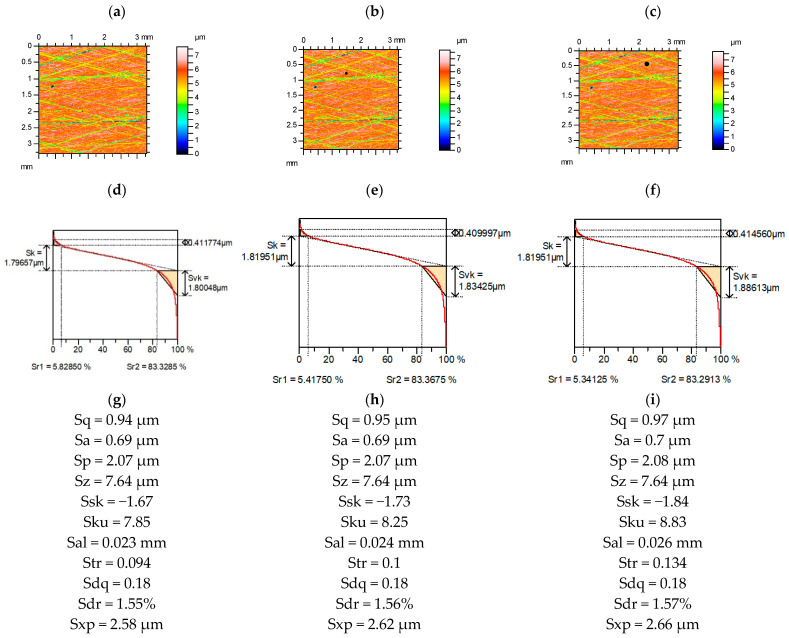
Contour plots (**a**–**c**), material ratio curves (**d**–**f**), and selected parameters (**g**–**i**) of the measured cylinder liner surface after plateau honing (**a**,**d**,**g**), with an additional circular valley with a diameter of 0.1 mm (**b**,**e**,**h**) and 0.15 mm (**c**,**f**,**i**).

**Figure 3 materials-15-04758-f003:**
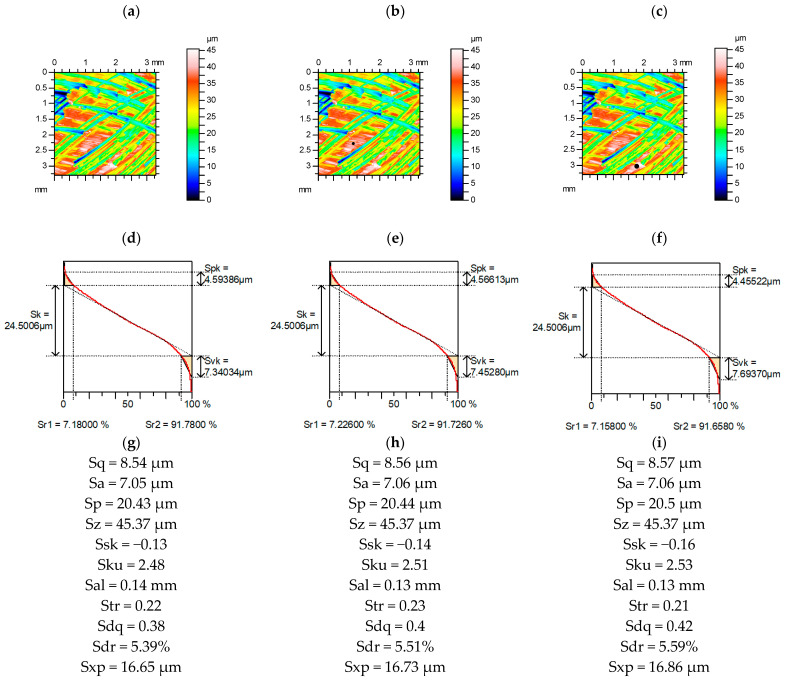
Contour plots (**a**–**c**), material ratio curves (**d**–**f**), and selected parameters (**g**–**i**) of the measured milled surface (**a**,**d**,**g**), with an additional circular valley with a diameter of 0.1 mm (**b**,**e**,**h**) and 0.15 mm (**c**,**f**,**i**).

**Figure 4 materials-15-04758-f004:**
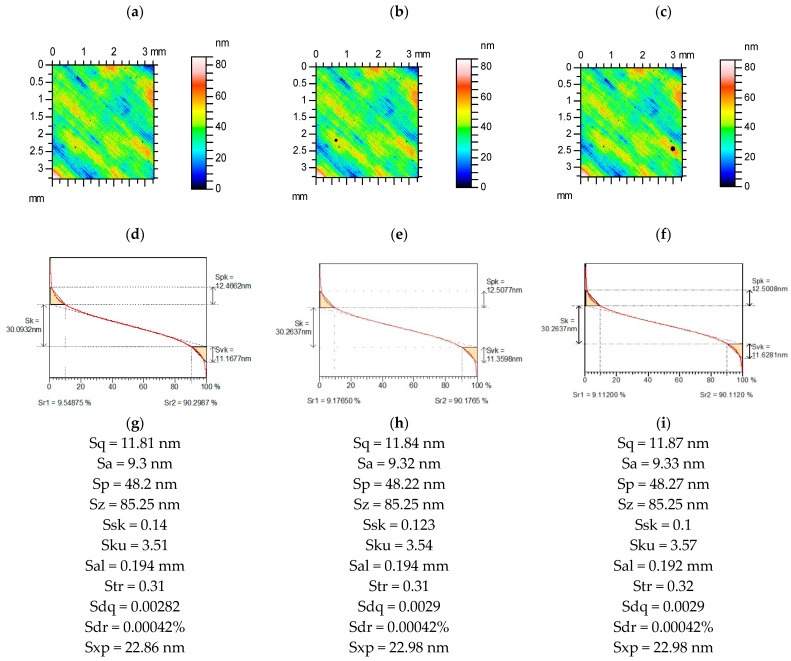
Contour plots (**a**–**c**), material ratio curves (**d**–**f**), and selected parameters (**g**–**i**) of the measured surface after polishing (**a**,**d**,**g**), with an additional circular valley with a diameter of 0.1 mm (**b**,**e**,**h**) and 0.15 mm (**c**,**f**,**i**).

**Figure 5 materials-15-04758-f005:**
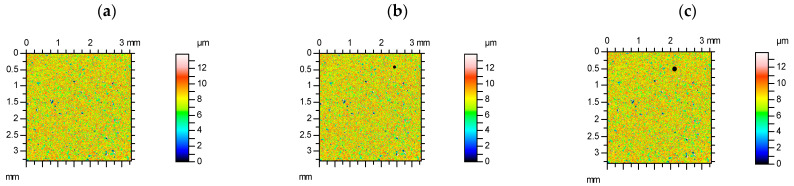
Contour plots (**a**–**c**), material ratio curves (**d**–**f**), and selected parameters (**g**–**i**) of the measured surface after abrasive blasting (**a**,**d**,**g**), with an additional circular valley with a diameter of 0.1 mm (**b**,**e**,**h**) and 0.15 mm (**c**,**f**,**i**).

**Figure 6 materials-15-04758-f006:**
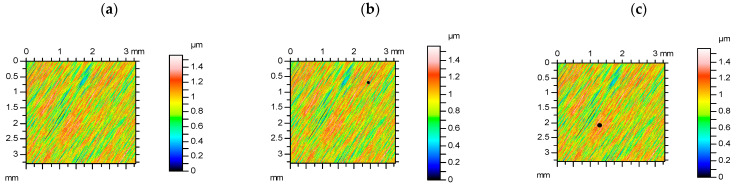
Contour plots (**a**–**c**), material ratio curves (**d**–**f**), and selected parameters (**g**–**i**) of the measured ground surface (**a**,**d**,**g**), with an additional circular valley with a diameter of 0.1 mm (**b**,**e**,**h**) and 0.15 mm (**c**,**f**,**i**).

**Figure 7 materials-15-04758-f007:**
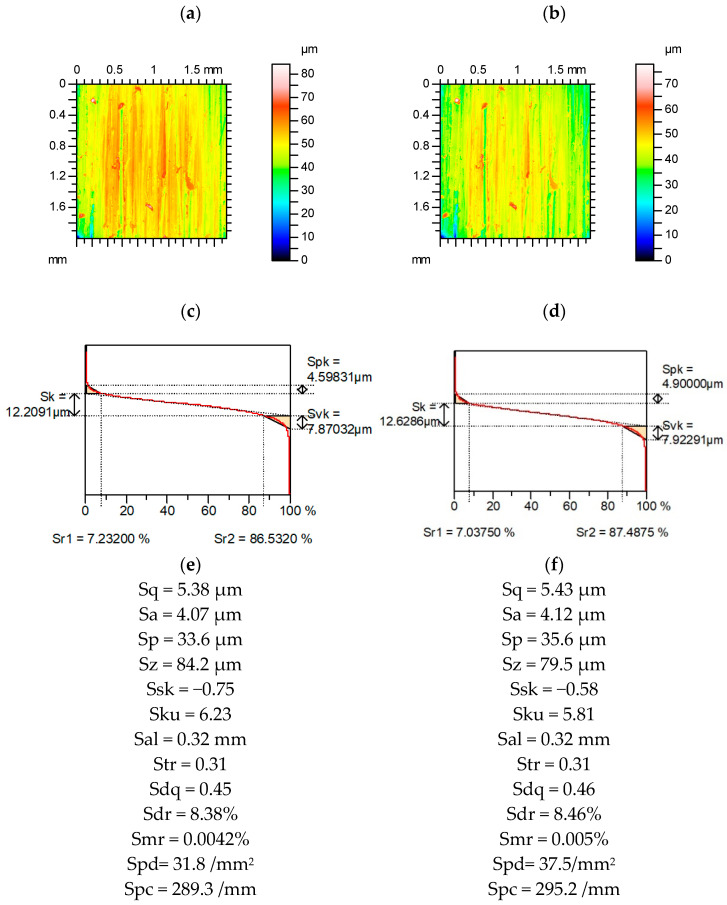
Contour plots (**a**,**b**), material ratio curves (**c**,**d**), and selected parameters (**e**,**f**) of the measured (**a**,**c**,**e**) and re-measured (**b**,**d**,**f**) ground surface.

**Figure 8 materials-15-04758-f008:**
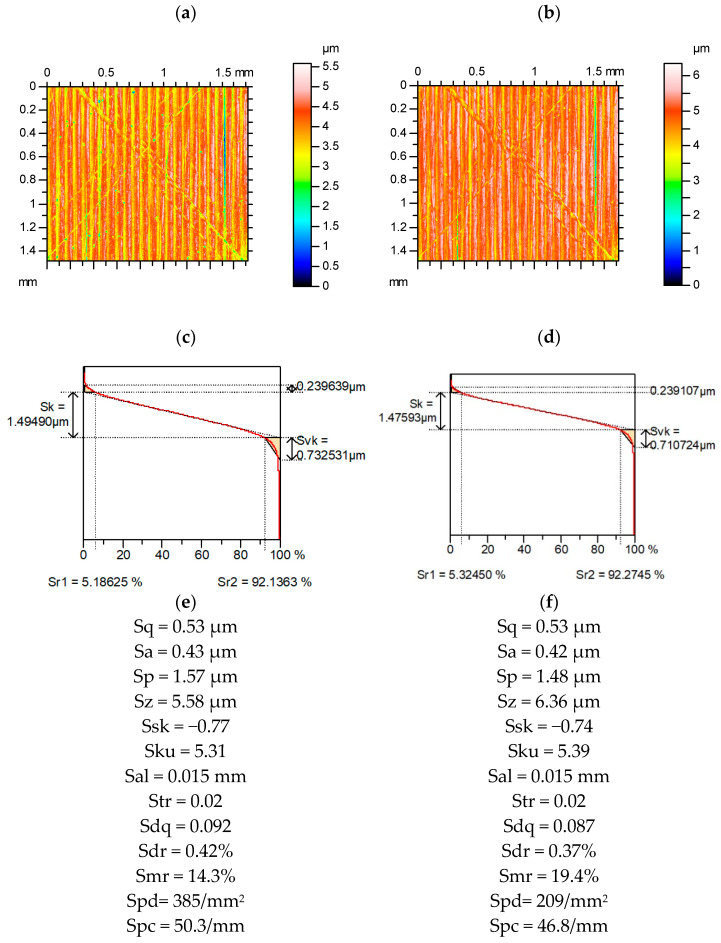
Contour plots (**a**,**b**), material ratio curves (**c**,**d**), and selected parameters (**e**,**f**) of the measured (**a**,**c**,**e**) and the re-measured (**b**,**d**,**f**) surface after lapping.

**Figure 9 materials-15-04758-f009:**
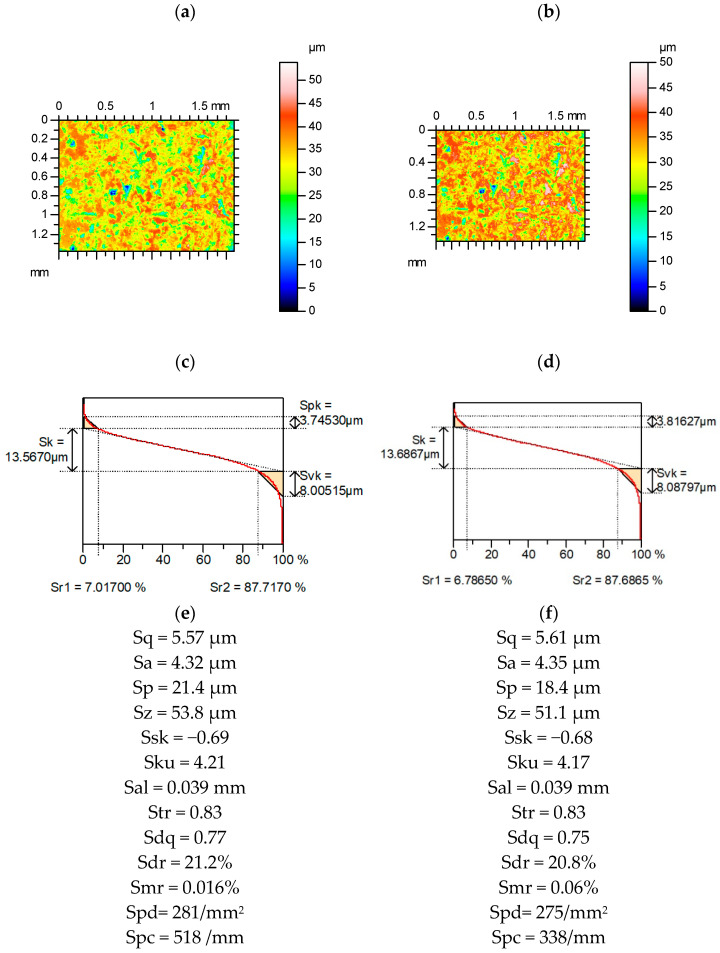
Contour plots (**a**,**b**), material ratio curves (**c**,**d**), and selected parameters (**e**,**f**) of the measured (**a**,**c**,**e**) and re-measured (**b**,**d**,**f**) surface after abrasive blasting.

**Figure 10 materials-15-04758-f010:**
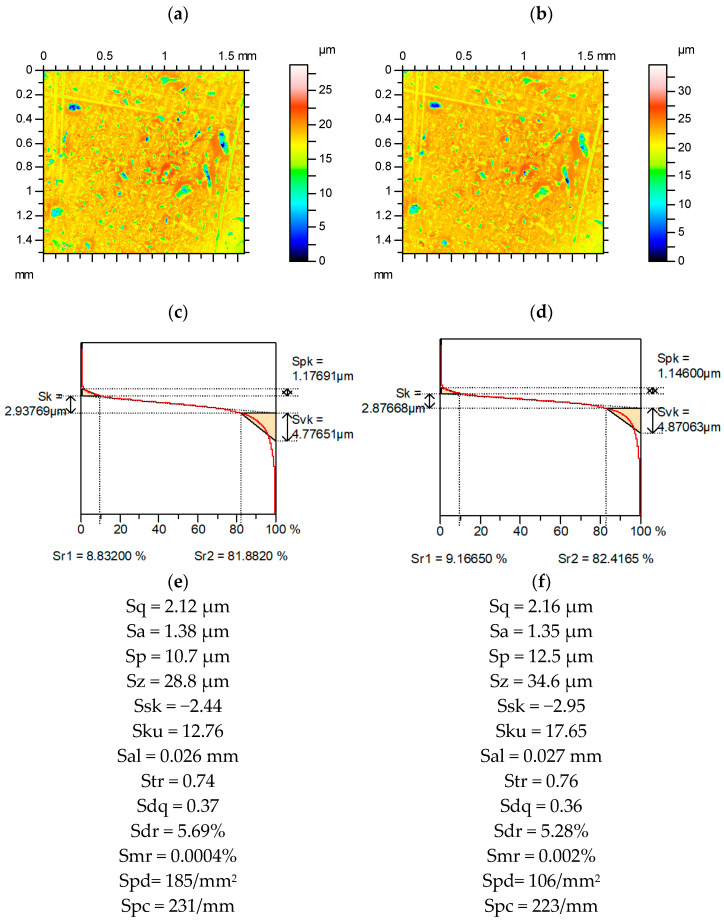
Contour plots (**a**,**b**), material ratio curves (**c**,**d**), and selected parameters (**e**,**f**) of the measured (**a**,**c**,**e**) and re-measured (**b**,**d**,**f**) surface after abrasive blasting and lapping.

**Figure 11 materials-15-04758-f011:**
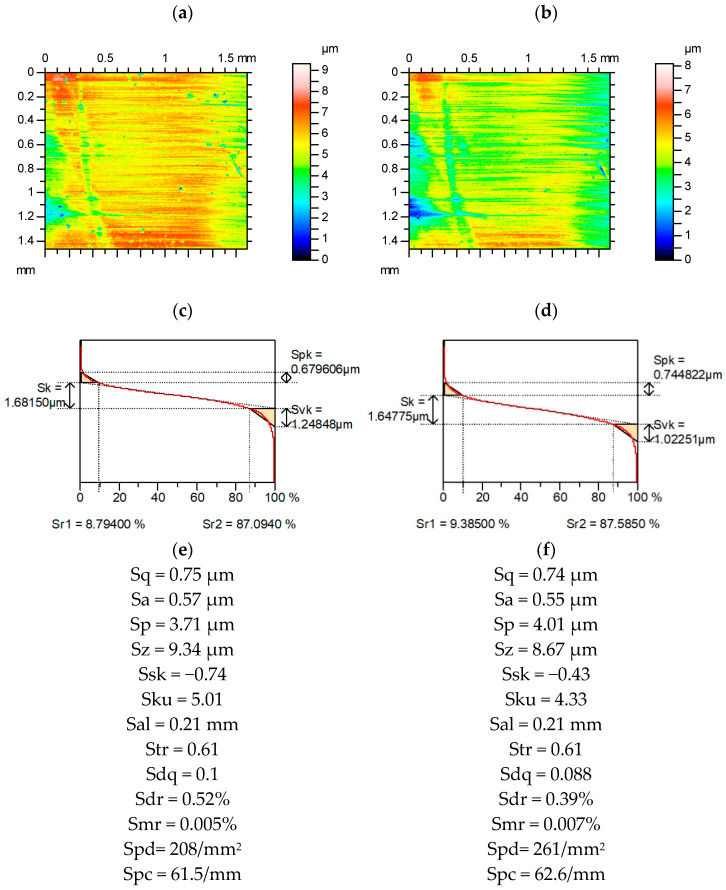
Contour plots (**a**,**b**), material ratio curves (**c**,**d**), and selected parameters (**e**,**f**) of the measured (**a**,**c**,**e**) and re-measured (**b**,**d**,**f**) ground surface.

## Data Availability

Not applicable.

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
