# Peer review of "The Effects of Selected Measurement Errors on Surface Texture Parameters"

_materials, 2022, doi:10.3390/ma15144758_

Round 1

Reviewer 1 Report

The manuscript is devoted to areal parameters of surface texture measurement and various types of errors that affect surface texture parameters.

The manuscript is clear, detailed, has a large number of drawings supporting the conclusions, and is presented in a well-structured manner. The citations in the manuscript are up-to-date.

The methods section provides description of methods and details.

The figures/tables/images/schemes are appropriate, clear and informative. 

The conclusions are based on the evidence and results presented in the manuscript.

Obtained results may be useful for further using in industry.

However, I would like to make a few remarks and ask a few questions:

1. It is not clear from the manuscript what the ultimate goal of this study is. I would like to suggest adding why you did this work.

2. What is the reason that you chose this particular value of areal parameters. What will happen if the surfaces with dramatically lower or higher texture (roughness) parameters will be analyzed?

3. Why you analyzed surfaces after 3 months? 

4. What we should do with the problem that results of measurement are varied? 

5. From my point of view, it would be better to add some recommendations how to use your results, and provide some explanation why it happen. 

Thank you for the opportunity to review the results of your research.

I enjoy doing this work. 

Author Response

Dear Reviewer,

Thank you very much for your valuable comments. We modified tje paper according to your suggestions.

Reviewer 2 Report

Very interesting research. The comments with the improvement are as follows:

1. A flowchart could be displayed at the beginning of the Materials and Methods section to make it easier to follow the methodology.

2. How was the sampling area selected (3.29 mm x 3.29 mm)?

3. The authors state the following: "Surfaces after various treatments, such as grinding, lapping, abrasive blasting, and abrasive blasting followed by lapping, were measured and analysed". It is not clear in the results which processing operations was applied where and when.

4. Further analyse and discuss the impact of processing operations (grinding, lapping, etc.) on the obtained results.

5. How the results and methodology can be applied in practice.

6. State in the conclusions the limitations of the methodology and future research.

Author Response

(The authors gave the same response as above.)

Round 2

Reviewer 2 Report

The manuscript has been corrected, so I suggest accepting it.